# A Ternary Copper (II) Complex with 4-Fluorophenoxyacetic Acid Hydrazide in Combination with Antibiotics Exhibits Positive Synergistic Effect against *Salmonella* Typhimurium

**DOI:** 10.3390/antibiotics11030388

**Published:** 2022-03-15

**Authors:** Guilherme Paz Monteiro, Roberta Torres de Melo, Micaela Guidotti-Takeuchi, Carolyne Ferreira Dumont, Rosanne Aparecida Capanema Ribeiro, Wendell Guerra, Luana Munique Sousa Ramos, Drielly Aparecida Paixão, Fernanda Aparecida Longato dos Santos, Dália dos Prazeres Rodrigues, Peter Boleij, Patrícia Giovana Hoepers, Daise Aparecida Rossi

**Affiliations:** 1Laboratory of Molecular Epidemiology, Federal University of Uberlândia, Uberlândia 38402-018, Brazil; guil.paz@hotmail.com (G.P.M.); micaelaguidotti@gmail.com (M.G.-T.); carolfdumont@gmail.com (C.F.D.); rosanneacr@yahoo.com.br (R.A.C.R.); fe.longato@hotmail.com (F.A.L.d.S.); daise.rossi@ufu.br (D.A.R.); 2Institute of Chemistry, Federal University of Uberlândia, Uberlândia 38402-018, Brazil; wendell.guerra@ufu.br (W.G.); luana_munique@hotmail.com (L.M.S.R.); paixaodrielly@hotmail.com (D.A.P.); 3Laboratory of Enterobacteriaceae, Oswaldo Cruz Foundation (FIOCRUZ), Rio de Janeiro 21040-900, Brazil; dalia.rodrigues@ioc.fiocruz.br; 4Check-Points B.V., 6709 PD Wageningen, The Netherlands; peterboleij@check-points.com; 5Postgraduate Program in Veterinary Science, Federal University of Uberlândia, Uberlândia 38402-018, Brazil; patriciag.hoepers@gmail.com

**Keywords:** food safety, multi drug resistance (MDR), salmonellosis

## Abstract

*Salmonella* spp. continues to figure prominently in world epidemiological registries as one of the leading causes of bacterial foodborne disease. We characterised 43 Brazilian lineages of *Salmonella* Typhimurium (ST) strains, characterized drug resistance patterns, tested copper (II) complex as control options, and proposed effective antimicrobial measures. The minimum inhibitory concentration was evaluated for seven antimicrobials, isolated and combined with the copper (II) complex [Cu(4-FH)(phen)(ClO_4_)_2_] (4-FH = 4-fluorophenoxyacetic acid hydrazide and phen = 1,10-phenanthroline), known as DRI-12, in planktonic and sessile ST. In parallel, 42 resistance genes were screened (PCR/microarray). All strains were multidrug resistant (MDR). Resistance to carbapenems and polymyxins (86 and 88%, respectively) have drawn attention to the emergence of the problem in Brazil, and resistance is observed also to CIP and CFT (42 and 67%, respectively), the drugs of choice in treatment. Resistance to beta-lactams was associated with the genes *bla*_TEM_/*bla*_CTX-M_ in 39% of the strains. Lower concentrations of DRI-12 (62.7 mg/L, or 100 μM) controlled planktonic and sessile ST in relation to AMP/SUL/TET and AMP/SUL/TET/COL, respectively. The synergistic effect provided by DRI-12 was significant for COL/CFT and COL/AMP in planktonic and sessile ST, respectively, and represents promising alternatives for the control of MDR ST.

## 1. Introduction

Foodborne disease-causing bacteria pose a major threat to public health worldwide. In particular, salmonellosis stands out as the fourth global cause of diarrheal disease and the second main foodborne disease affecting Europe, the USA and Brazil [1,2,3,4]. In humans, salmonellosis can cause diseases with different levels of severity, being commonly a self-limiting gastroenteritis without the need for antibiotic therapy, with symptoms such as diarrhoea, fever and abdominal pain that occurs from 12 to 72 h after consumption of contaminated food. However, more complex cases with extraintestinal infection or septicaemia may occur and need antimicrobial treatment, such as fluoroquinolones and ceftriaxone (CFT) [1].

The most incriminated foods in the transmission of human salmonellosis are poultry meats and their derivatives, but there is also frequent involvement of foods such as contaminated beef and pork, vegetables, milk, and water [5]. During the different stages of food processing, from animal production to consumption in the domestic environment, these products are susceptible to cross-contamination, which represents the main risk factor for increased rates of foodborne infection [6]. *S.* Typhimurium (ST) is one of the most involved serovars in foodborne outbreaks [5]; in Brazil, it stands out for having been reported as the main isolated serovar from systemic infections in humans [7] and, in 2020, the third-most identified in chicken slaughterhouses by the federal inspection service [8]. In the European Union, in 2019, it was the fourth-most reported serovar in outbreaks [2], and the third-most in the United States [5].

The emergence of problems related to this pathogen has increased significantly due to its emerging antimicrobial resistance and its inherent ability to adhere to surfaces and, consequently, form biofilms [9]. The genus *Salmonella* belongs to a group of bacteria of global importance regarding their high levels of antimicrobial resistance. Throughout its evolution, *Salmonella* has developed a range of molecular mechanisms of drug resistance at the level of both global and local transcriptional regulators, such as membrane and/or cytosolic sensors and efflux pumps, which intensify the high number of multidrug-resistant (MDR) strains [10]. The biofilm formation is an essential factor in the maintenance and greater adaptation of the microorganism, and it has an impact on the contamination of food during its processing and on the potential acquisition of virulence and resistance factors. In general, the basic stages of sessile structure formation include cell adhesion to surfaces and biofilm maturation, combined with the secretion of a protective extracellular matrix and cell dispersion [11].

Given this scenario, the need to create new strategies to control this microorganism is paramount. Among the most recent approaches, those using metal-based drugs can be very interesting [12,13]. In this sense, copper complexes have received attention due to their promising results as antibacterial agents [14,15]. For example, copper (II) complexes with N(4)-ortho, N(4)-meta and N(4)-para-tolyl thiosemicarbazones derived from 2-formyl and 2-acetylpyridine were tested against *Salmonella* Typhimurium and exhibited MIC values between 5–30 μmol.L^−1^, while the drug chloramphenicol presented an MIC value of 12 μmol.L^−1^ [16]. These observations suggest that copper complexes may be useful against this bacterial class. Moreover, copper is an essential trace element with importance for the functioning of several enzymes involved in energy metabolism, respiration and DNA synthesis in the cell, which make it very attractive for the development of safe and effective drugs [17,18,19]. Thus, DRI-12 was recently prepared [20] as a selective copper (II) complex containing 4-fluorophenoxyacetic acid hydrazide and 1,10-phenanthroline, and it has shown DNA-cleaving and pro-apoptotic properties in cancer cells, as well as selectivity and non-mutagenicity in tests with *Drosophila melanogaster* [21,22].

Thus, our approach included characterising the resistance profiles of ST to antimicrobials of importance in human and veterinary medicine in strains of ST isolated from food and human patients in Brazil between 2011 and 2017. In parallel, this study evaluated the isolated effect of the copper (II) complex called (DRI-12) [20] in the control of MDR strains in planktonic and sessile forms.

## 2. Results

### 2.1. Antimicrobial Resistance

The percentages of resistance and/or intermediate resistance measured for ST are described in Table 1. We observed the highest resistances for SUL, COL and MER, with 100, 88 and 86% resistance, respectively. CIP and TET were the most effective antimicrobials in controlling ST, and the resistance for both was 42% in the tested strains. Low concentrations were sufficient to achieve MIC_50_ for CIP (0.0312 μg/mL) and TET (<0.5 μg/mL), whereas for AMP, MER and COL the values were 32, 4 and 4 μg/mL, respectively, while for SUL the concentration reached >2048 μg/mL. MIC_90_ was superior to the breakpoint for all drugs except for CIP (0.25 μg/mL). For DRI-12, the MIC_50_ and MIC_90_ values were equivalent (62.68 μg/mL) and lower than the concentrations of AMP, TET and SUL necessary for the 90% inhibition of ST strains. 

A total of 28 phenotypic resistance profiles were obtained, varying according to resistance and/or intermediate resistance to each antimicrobial, which determined the MDR pattern for all STs according to the criterion of Magiorakos et al. [23] (Appendix A). There was no significant difference in the prevalence of each of the profiles according to the origin of the strains (*p* < 0.05, Fisher Test), but profiles with pentaresistance or higher represented a significant majority of the strains (29/43, or 67.4%; *p* < 0.05), more so even than in strains from humans (19/29—65.5%; *p* = 0.0483). Joint resistance to SUL, COL and MER was identified in 21/28 (75%) profiles, representing 32/43 (74.4%) strains. Profiles that demonstrated resistance to CIP and CFT, concomitantly or not, represented 39/43 (90.7%) strains.

As for genotypic resistance evaluated by conventional PCR and microarray (Appendix A), we observed that no gene linked to resistance to fluoroquinolones, carbapenems or AMPC was identified in ST. As for the genes encoding beta-lactamases, we identified only *bla*_TEM_ and *bla*_CTX-M-2_, found in 17/43 (39.5%) and 1/43 (2.3%) isolates, respectively. The *bla*_TEM_-positive STs comprised eight strains isolated from food and nine from humans, which expressed phenotypic resistance to ampicillin (AMP) (17/17, 100%). There is extensive spread of *bla*_TEM_, which was the most detected gene and is considered dominant in human feces samples at a rate of 88.8% (8/9). A noteworthy observation for food samples shows that this gene was most prevalent (7/8—87.5%) in poultry products and swine samples. These results amplify resistance to the penicillin class, as shown in the evaluation in Appendix A, of which 75% (21/28) of the resistance profiles indicated resistance to ampicillin. The positive strain for *bla*_CTX-M-2_ from food showed concomitant resistance to AMP, CFT and MER, in addition to resistance to standard first-line therapy for *Salmonella* infections, SUL and CIP (profile XIII—Appendix A).

### 2.2. Copper Complex Synergism in Antimicrobials

In all synergism analyses (Figure 1), we observed either maintenance or reduction in MIC for planktonic ST control.

The synergism between AMP, TET, MER and CIP associated with DRI-12 was not detected in the planktonic form, as the MIC values remained constant in both the control and test (Md = 128, 64, 0.125 and 0.125 μg/mL, respectively). Despite this, one strain of ST showed a reduction in MIC for TET (2-fold), MER (≥64-fold) and CIP (2-fold), demonstrating that the synergistic effect is strain-dependent. For SUL, there was a reduction in the MIC value in the presence of DRI-12 (median change from ≥4096 to 256 μg/mL, 16-fold reduction), evident in 3/5 strains, but this reduction was not significant (*p* > 0.05, Mann–Whitney). For COL and CFT, we detected that the MIC reduction was significant (*p* = 0.0159 and 0.0476, respectively, Mann–Whitney), evident in all strains, and determined a median change from 8 to 1 μg/mL (8-fold reduction) and from 2 to ≥0.0625 μg/mL (32-fold reduction), respectively, which made it possible to reclassify the strains’ resistance and intermediate resistance profiles to “susceptible” to these drugs after the inclusion of [Cu(4-FH)(phen)(ClO_4_)_2_].

High concentrations of the tested drugs were needed in isolation to control the structure of the biofilms of the five ST strains tested. The median values (maximum and minimum) for AMP, CFT, CIP, COL, MER, SUL and TET were, respectively, ≥2048 μg/mL (2048 and 512 μg/mL), 64 μg/mL (256 and 1 μg/mL), 2 μg/mL (4 and 0.25 μg/mL), ≥1024 μg/mL (1024 and 512 μg/mL), 16 μg/mL (512 and 2 μg/mL), ≥16,384 μg/mL (idem) and 256 μg/mL (512 and 128 μg/mL). For the isolated copper complex, concentrations equivalent to 50 μM (31.34 μg/mL) (1/5, 20%), 100 μM (62.68 μg/mL) (3/5, 60%) and 200 μM (125.36 μg/mL) (1/5, 20%) were required for the inhibition of biofilm ST cells, with a median equivalent to 100 mM (62.68 μg/mL), a concentration lower than that required for control using AMP, COL, SUL and TET.

The synergistic effect of using DRI-12 was detected for all antimicrobial drugs tested in at least 2/5 (40%) of the biofilm ST strains. For SUL and CIP, despite the presence of 2/5 strains with reductions of ≥4096 and 2-fold, respectively, in the MIC value, the median of the control and test groups remained equivalent (16,384 and 2 μg/mL), showing the absence of a statistical difference (*p* > 0.05, Mann–Whitney). Although there was no significant difference, we observed in 3/5 (60%) of the biolfim ST strains a reduction in MIC, and a change in the median of MIC from 16 to 2 μg/mL (8-fold reduction) for MER supplemented with DRI-12. This made it possible to change the general profile of the five ST strains from resistant to intermediate. Exposure to TET and CFT added to DRI-12 promoted a change in MIC in 4/5 (80%) of the ST strains in biofilms, reducing the medians from 256 to 64 μg/mL (4-fold reduction) and from 64 to 2 μg/mL (32-fold reduction) (*p* > 0.05, Mann–Whitney). For AMP and COL, the MIC reduction was significant (*p* = 0.0159 and *p* = 0.0079, respectively, Mann–Whitney), identified in 4/5 (80%) and 5/5 (100%) of the biofilm ST strains, and promoted a reduction in the median by 8- and 64-fold, respectively (Md_AMP_ ≥ 2048 for 256 μg/mL; Md_COL_ ≥ 1024 for 16 μg/mL).

### 2.3. Image Analysis of the ST Biofilms

In the SEM, we observed the formation of mature biofilms with an evident three-dimensional matrix structure, as well as the organization of sessile cells (Figure 2).

In Figure 2A,B (control), we can see that the biofilms did not reach the maximum degree of maturity in the tests since the stable structure of the matrix was not fully formed and there were many dispersed microcolonies with the presence of visible filaments of matrix in their outermost part, but they showed considerable thickness for the protection of cells inside the biofilm. The tests with the copper complex (Figure 2C,D) showed the loss of the three-dimensional ultrastructure of the biofilms, with exposed cells and cells undergoing cell death or disintegration and destruction of the matrix, in addition to fragmentation of the biomass, which is also visible to the naked eye in assays performed on microplates.

## 3. Discussion

### 3.1. Antibiotic Susceptibility

Our findings demonstrated a high percentage of resistance or intermediate resistance by the strains to SUL, COL and MER, the commonly used drugs in veterinary and human medicine in Brazil.

A global study carried out in 2019 showed that for sulfonamides, as well as tetracyclines (TET) and penicillins, ST isolates from poultry and swine animal production have the highest resistance rates. Consequently, Brazil has emerged as one of the main emerging hotspots for the maintenance of these strains [25]. Unanimous resistance to sulfonamides has been widely reported in ST isolated from humans, seawater and animal products [26], and is related to the acquisition of the *sul1* and *sul2* genes responsible for the production of dihydropteroate synthase [27], as well as *sul3* acquired via plasmid [28]. These resistant patterns play a critical role in modifying antibiotic resistance in pathogenic bacteria that directly affect human health upon entering the food chain [29]. In Portugal, a study of 200 strains, including ST, investigated the spread of the three resistance genes, with worrying results (76%—*sul1*; 37%—*sul2*; and 7%—*sul3*), mainly related to swine food products [26]. In Morocco, a 25% resistance to sulfonamides was identified in *Salmonella* isolated from different products to be marketed in the food industry (cheese, milk, meat, sausage and chicken meat) [30]. Even more alarming results were reported by Khallaf et al., also in Morocco, of 64% resistance [31], and by Proroga et al. in Italy, with 69% resistance [32].

For meropenem (MER), the high resistance pattern of ST strains signals a public health concern. Considered a modern drug and one of the last alternatives for the treatment of super-resistant strains, MER is a beta-lactam belonging to the carbapenem subclass [33] with a mechanism of action in the inhibition of peptidoglycan synthesis [34]. In Brazil, the evaluation of the efficacy of drugs in the same class against different *Salmonella* serotypes has not shown resistance [33,34,35]. Another more recent study, also carried out in Brazil with *Salmonella* Heidelberg, identified resistance to MER in 25% of the strains [36].

In addition to MER, colistin (COL) (a class of polymyxins) is considered one of the last alternatives in the treatment of infections caused by MDR microorganisms, and is, therefore, considered critically important by the World Health Organization. High rates of resistance to this drug were also observed by other researchers, for example, in isolates of *S.* Enteritidis isolated from broilers in Ecuador (100%) [37]; in human clinical samples in Brazil, demonstrating a time trend for increased resistance to COL [38]; and in *S.* Heidelberg isolated from food (100%) [36]. One of the hypotheses for the high percentage of COL-resistant strains is that this phenotype may have emerged through selection pressure. Until 2016, this antimicrobial was used in Brazil as a growth promoter in animal production and its use for this purpose only ceased after its prohibition by the Ministry of Agriculture, Livestock and Supply (MAPA), in line with international action and with the first reports of COL resistance in the world [39]. The high rates of resistance reported in this and other mentioned studies may reflect the consequences of the long-term use of COL in animal production.

Although CIP demonstrates the greatest effectiveness for the control of more than half of the strains (25 ST), the rate of 41.9% ST resistance (1/18) or intermediate resistance (17/18) is worrisome, as this is the drug of choice for the treatment of human salmonellosis [4]. Lin et al. [40] noted the trend of increasing resistance to this antimicrobial in Asia, where they measured 39% resistance after analysing 82 strains of different serovars. The use of CIP as a growth promoter has been prohibited in Brazilian animal production since 2009, being allowed only for therapeutic use [41]. Despite this, the number of *Salmonella* resistant to this drug have also been increasing in Brazil. Mendonça et al. [37] found 7% resistance in 111 strains of *S.* Enteritidis isolated from chicken meat; Melo et al. [30] detected 10% resistance in *S.* Heidelberg; and Pribul et al. [42] observed 42.6% resistance, with serovar ST being the most associated with this characteristic.

Similar to what was observed for CIP, TET was also the most effective in controlling 25/43 (58.1%) strains of ST. Nevertheless, this result indicates a significant percentage of resistant strains. A study carried out by Gargano et al. [43] in Italy with *Salmonella* from different foods of animal origin identified a resistance of 33.3% to TET. In Mexico, resistance was 40.3% in ST isolated from beef [44]. The mechanisms involved in TET resistance allow great diversification of resistance acquisition (*tet* A–E and *tet* G–J) [45] and include more than sixty described genes [46] that are determined by plasmid acquisition [47], through the action of efflux pumps [48], by encoding ribosomal protection proteins (*tet* M, *tet* O, *tet* W and *tet* Q) [46] and by enzymatic inactivation (*tet* X) [49]. This impact is amplified when its bioaccumulation in wastewater is evaluated [50], the abundance of which can maximize the horizontal transfer of resistance genes [51].

For CFT, we identified resistance in 10/43 (23.2%) strains and intermediate resistance in 19/43 (44.2%) of the strains. This drug belongs to the class of beta-lactams and the subclass of third generation cephalosporins, being used mainly for the treatment of children, the elderly and people with compromised immune systems [52]. It is also considered critically important by the WHO [53] for its use in more severe cases of salmonellosis. According to Carattoli et al. [34] and Carson et al. [54], CFT is one of the cephalosporins to which *Salmonella* has the highest resistance rate, and this is considered to be a consequence of the large-scale use of ceftiofur, a drug for veterinary use [55], which ended up culminating in cross-resistance to CFT, as the two are drugs belonging to the same subclass [56,57].

The prevalence of resistance to AMP (51%) found in our study corroborates other studies carried out in Brazil in the area of veterinary and human medicine that identified 87% resistance to this drug in *Salmonella* isolates from chicken carcasses [58], and in 63% of 11.447 strains of *S.* Typhimurium from the entire production process of the meat products chain [59]. AMP resistance in ST is expected, according to a retrospective study that analyzed samples from different food sources, animals and humans over a period of more than 50 years. The authors pointed out that this characteristic originated in this serovars, several years earlier than reported, through clonal expansions that determined the multiple plasmid acquisitions [60]. In parallel, data from the European Union and the USA have shown the involvement of the indiscriminate use of this class in the development of acquired resistance, since antibiotics of the penicillin class account for about 31.1% of sales of drugs for use in farm animals marketed in 31 European countries, and 13% of sales in the USA [61,62]. This is because the drug is considered to be the first penicillin active against Enterobacteriaceae [60].

Considering that all strains showed resistance to at least one of the beta-lactam drugs (AMP, CFT and MER), in 17 of them (39.5%), this profile can be explained by the presence and expression of one of the *bla*_TEM_ or *bla*_CTX-M_ genes. Our study reflects the important prevalence of these two genes as also reported in different studies carried out in Korea in poultry products (72.7% *bla*_TEM_ and 18.1% *bla*_CTX-M_) [63]; in clinical isolates (4.47% for *bla*_TEM_ [64] and 3.38% for *bla*_CTX-M_ [65]); and in beef products (14.5% *bla*_TEM_ and 28% *bla*_CTX-M_) in ST [66].

In general, the current study detected a high prevalence of β-lactamase resistance genes in *Salmonella* isolates. These results align with the increasing rates of antimicrobial resistance and the production of extended-spectrum β-lactamase (ESBL) [67], highlighting caution in the use of these antibiotics that exacerbate the resistance of *Salmonella* isolates to three important subclasses of β-lactams and compromise first-line antimicrobial therapy [59,68].

The diversity of resistance profiles identified in our study, combined with an MDR pattern in all strains and the statistical significance of a greater number of ST strains that are pentaresistance or higher, makes the problem of antimicrobial resistance in our country more evident. A 2012 survey by the National Health Surveillance Agency (ANVISA) determined that among 250 strains, ST was the serovar with the second-highest prevalence of MDR (28%), showing resistance to at least five different antimicrobials, surpassed only by *S.* Enteritidis (30%) [69]. In parallel, a 21 year analysis in the US showed the involvement of serovar ST with pentaresistance or higher, in addition to the correlation between MDR and human samples [59].

### 3.2. Copper Complex Effect

The copper complex [Cu(4-FH)(phen)(ClO_4_)_2_] (DRI-12) was previously evaluated for its physicochemical properties, cytotoxicity in tumours [21], selectivity and non-mutagenicity in tests with *Drosophila melanogaster* (in press) [22], as well as its antimycobacterial activity [20]. These studies revealed that DRI-12 has high potency and low toxicity, which makes it a promising compound to fight cancer and microorganisms. Additionally, previous studies revealed that copper complexes show effectiveness in antimicrobial control in hospital environments [70] upon inclusion in surfaces with bactericidal properties [71], demonstrating the extensive dynamics of applicability for pathogen control.

In Gram-negative strains, previous studies with other copper compounds have demonstrated moderate-to-weak antibacterial properties, with MIC_95_ assuming values greater than 125 μM [72], reaching up to 500 ppm in the control of *E. coli* isolated from bovine mastitis [73], and with MIC ranging from 100 to 200 μg/mL and MBC reaching 1600 μg/mL in MDR nosocomial pathogens [74]. However, our study found a superior efficacy of DRI-12 alone, with MIC_90_ values equivalent to the median value required for biofilm control, equal to 100 μM (62.68 ppm/62.68 μg/mL).

Although with important chemical differences, other metals compounds reviewed in the literature were less effective for *Salmonella* control, with minimum inhibitory concentration (MIC) values between 5000 and 2500 μg/mL for titanium dioxide (TiO_2_) loaded nanoparticles [75] and up to 78 uM for a silver nitrate (AgNO_3_) solution in the control of planktonic forms of multidrug-resistant ST isolated from calves with diarrhoea [76]. In addition, for bulk zinc oxide (ZnO), the level of inhibition for five *Salmonella* strains was higher (MIC between 312.5 to 625 μg/mL) than that reported for nanoparticles [77,78].

In our study, the combination of antimicrobials from the polymyxin (COL) and beta-lactam (AMP and CFT) classes associated with DRI-12 had a significant effect, directly impacting planktonic forms and biofilm formation. It is peculiar to note that the three antimicrobials that act directly on the cell wall acted effectively when associated with the inherent mechanisms of copper.

Polymyxins are polycationic peptides that act initially by binding to the phosphate group of the LPS of the outer membrane, which promotes the displacement of Mg^2+^ and Ca^2+^ ions that help to bridge the bonds between adjacent LPS molecules, leading to their rupture. Consequently, the permeability of the cytoplasmic membrane is modified, leading to a loss of cell content, lysis and death [79]. The bactericidal activity of COL is independent of its passage through the microbial cell [80], which makes it possible to maximize its mode of action via recently developed approaches, such as the use of copper compounds (DRI-12) which interact with the negatively charged membrane of microorganisms [81]. This results in synergistic interactions that allow the reversion of resistance on planktonic bacteria and expressive results for the control of biofilms. This phenomenon explains the inhibition of microbial growth by DRI-12 alone in more than 80% of the ST strains in the planktonic form. In association, COL + DRI-12 likely promotes a dual effect on the microbial plasma membrane through oxidative action and binding with the outer membrane by copper, allowing a more effective action of COL on LPS.

Beta-lactams such as AMP and CFT have the inhibition of microbial cell wall synthesis as their first potential target. The mechanisms of resistance to these drugs are linked to the production of β-lactamases that directly impact classes of penicillins, cephalosporins and carbapenems [82]. The class of penicillins that bind to the primary receptor is known as membrane-bound penicillin-binding proteins (PBPs), which disrupt the third and final stage of bacterial cell wall synthesis, leading to cell lysis [83]. Synergistic combinations dependent on metal ions, such as copper and AMP, are currently being proposed as novel strategies for re-sensitization of AMR strains, including the *Enterobacteriacea* family, which exhibit promising effects when compared to antibiotic treatment alone [84]. The sensitizing effect of transition metal ions is primarily directed to the destabilization of the outer membrane through the interaction with electronegative chemical groups that promote an increase in permeability and interruption of internal processes, allowing greater effect and antibiotic activity [85].

CFT is a very closely related to penicillin in its mode of action and binds to PBPs, inhibiting mucopeptide synthesis in the bacterial cell wall. Furthermore, CFT promotes the formation of defective cell walls due to its binding with important enzymes (carboxypeptidases, endopeptidases and transpeptidases) responsible for cell wall synthesis [86]. Metal complexes with CFT have pharmacological and antibacterial properties that have already been reported, and the result of this association can promote a synergistic effect that plays a key role in inhibiting bacterial growth. Although this mechanism is strain-dependent [87], Cu(II) ions appear to bind to nearby enzymes in a manner that is non-competitive to other mechanisms [88]. The result of this interaction with the microbial cell wall can cause damage by different modes of action, as with COL, potentiating the action of CFT [89].

In biofilms, the addition of DRI-12 promoted greater destruction of the biomass, likely due to matrix damage and exposure of sessile cells. Furthermore, this association may have maximized the action of the copper complex in the production of reactive oxygen species [90], causing the death of the outermost layer of the biofilm and allowing the destruction of the matrix and effective action of AMP and COL.

## 4. Materials and Methods

### 4.1. Origin of Strains and Study Design

The study was conducted with 43 strains of ST from food samples (20) and humans with salmonellosis (23), isolated between 2011 and 2017 (Appendix A). The strains are part of the Biological Collection of Bacteria of Interest in Public Health at FIOCRUZ (Instituto Oswaldo Cruz-RJ), which kindly provided the isolates and their origin information (Appendix A). The strains were screened for 42 genes related to antimicrobial resistance and phenotypic susceptibility to seven drugs representing seven important antimicrobial classes in human and veterinary medicine. In parallel, the strains were comparatively analysed for exposure to [Cu(4-FH)(phen)(ClO_4_)_2_], alone and in addition to the tested antimicrobials. The synergy between the chemical agents was verified in the planktonic and sessile forms of ST. The results were used to determine resistance profiles in addition to establishing a control proposal in multi-resistant strains.

### 4.2. Research of Genes Associated with Antimicrobial Resistance

The PCR technique was used to evaluate a panel composed of six genes associated with resistance (*bla*_TEM_, *bla*_SHV_, *bla*_CTX-M_ and *bla*_CMY-2_, classed as β-lactams, and *qnrS* and *qnrA*, classed as fluoroquinolones) (Table 2).

The genes evaluated were selected based on previous studies [91,92,93,94,95,96,97,98] related to resistance to antimicrobials belonging to the classes most used in human and veterinary medicine.

DNA extraction was performed using the Wizard Genomic Purification kit (Promega, Madison, WI, USA) according to the manufacturer’s instructions, and subsequently quantified in a Nanodrop spectrophotometer (Thermo Scientific**^®^**, Waltham, MA, USA). For the PCR reaction, 10 ng/μL of the DNA template was amplified in a thermocycler (Eppendorf**^®^**, Hamburg, Germany) using the GoTaq**^®^** Green Master Mix kit (Promega, Madison, WI, USA), with forward and reverse primer pairs (Table 1) in the reaction specific for each gene and ultra-pure water in a final volume of 25 μL. The *S.* Enteritidis ATCC 13076 strain was used as a positive control.

In all reactions, the initial denaturation occurred at 94 °C for 5 min and the final extension at 72 °C for 10 min, varying the conditions of temperature, time and number of cycle repetitions for each of the genes (Table 3). After amplification, the samples were electrophoresed on a 1.5% agarose gel immersed in 0.5× TBE buffer (Invitrogen**^®^**, Waltham, MA, USA) and stained with Syber Safe (Invitrogen**^®^**, Waltham, MA, USA). The 100 bp marker (Invitrogen**^®^**, Waltham, MA, USA) was used as a molecular weight reference standard, and the resulting bands after the run were visualized on the transilluminator under ultraviolet light (Loccus Biotecnologia**^®^**, Cotia, Brazil). In parallel, molecular microarray analysis was used to identify 42 specific genes of importance to the genus *Salmonella* related to drug resistance. We used the commercial kit CT103-XL Check MDR (Check-Points B.V., Wageningen, The Netherlands).

The DNA extraction was performed with the DNeasy Blood and Tissue kit (Qiagen, Hilden, Germany), adapting the manufacture’s protocol. The starting material was a pure colony isolated in TSA agar. In a 1.5 mL microtube, we added 180 μL of ATL buffer and 20 μL of proteinase K solution. Three colonies were resuspended in this solution, mixed well by vortexing and then incubated at 56 °C in a heating block for 60 min. The remaining steps were carried out according to the manufacturer’s protocol. The final eluted volume was diluted in a 1:5 ratio and used as the working solution. The 42 genes that were searched are described in Table 3. The Check-MDR-CT103XL (Check-Points B.V., Wageningen, The Netherlands) commercial kit consisted of three steps (DNA recognition, amplification and detection) and was used according to the manufacturer’s recommendations. In short, the microarray technique is based on a highly specific recognition of DNA sequences by a series of probes, followed by subsequent amplification of the ones that matched and paired with the DNA targets. Each DNA target is recognized by one specific probe that has in its structure a unique segment named by the manufacture as ZIP codes, which addresses the probes to a specific and unique position of hybridization on the microarray. Only the probes that perfectly matched with the DNA target will be amplified, and those that have at least one different nucleotide will not result in amplification products. After the PCR reaction, the amplified products are hybridized to the microarray present at the bottom of the tube, which later will be revealed by a colorimetric reaction and visualized in a specific tube reader. The reader connected to a computer shows an array image with a pattern of dots which are interpreted by Check-Points’ software, yielding the results.

### 4.3. Minimum Inhibitory Concentration (MIC)

The MIC was tested for seven antimicrobials belonging to seven classes and/or subclasses considered to be of critical importance by the World Health Organization, and that have been indicated for the treatment of human salmonellosis [53], namely: AMP (Vetnil, penicillin class—subclass of aminopenicillins), CFT (Triaxon, TEUTO, Anápolis, Brazil—class of β-lactams, subclass of third-generation cephalosporins), MER (meropenem, TEUTO, Anápolis, Brazil—class of β-lactams, subclass of carba-penems), CIP (Ciprodez, Biovet, Vargem Grande Paulista, Brazil—class of fluoroquinolones), COL (Colis-tek, Opem Pharmaceuticals, Consolação, Brazil—class of polymyxins, subclass of lipopeptides), SUL (sulfonamide, Sigma-Aldrich, St. Louis, USA—class of sulfonamides) and TET (tetracycline, Sigma-Aldrich, St. Louis, USA—class of tetracyclines). In addition, the test was also performed with the copper complex [Cu(4-FH)(phen)(ClO_4_)_2_].

The MIC was determined by the microdilution technique according to the protocol of the Clinical and Laboratory Standards Institute, and the classification of the strains as resistant, intermediate and sensitive also followed the clinical breakpoint guide for bacteria belonging to the order Enterobacterales [18]. For the copper complex, the MICs were compared to values obtained for commercial drugs.

Briefly, a standardized bacterial suspension was prepared at a concentration corresponding to 0.5 on the MacFarland scale, and eight concentrations of antibiotics were used, covering values above and below the breakpoints [20]. The maximum concentrations were 64.0, 16.0, 1.0, 32.0, 32.0, 2048.0, 64.0 and 62.7 μg/mL for AMP, CFT, CIP, MER, COL, SUL, TET and [Cu(4-FH)(phen)(ClO_4_)_2_], respectively.

Afterwards, the bacterial suspension was inoculated, and the microplates were incubated at 36 °C for 16–20 h. The reading was performed visually, with the determination of MIC corresponding to the lowest concentration where no turbidity was observed. For classification as resistant, specific breakpoints were used for AMP (≥32 mg/L), CFT, COL, MER (≥4 mg/L), CIP (≥1 mg/L), SUL (≥12 mg/L) and TET (≥16 mg/L) [18].

The MDR character of the strains was determined using the criterion of Magiorakos et al. [17], which defines MDR as resistance or intermediate resistance to at least one drug belonging to three or more categories of antimicrobials, whether they are classes or subclasses.

In all tests, the *E. coli* ATCC 25922 strain was used as a positive control, and in the case of COL, a second positive control, *E. coli* NCTC 13846, was added, which is positive for the *mcr-1* gene associated with resistance to this drug.

### 4.4. The Synergistic Effect of the Copper Complex on the Planktonic and Sessile Forms of ST

The copper complex [Cu(4-fh)(phen)(ClO_4_)_2_] (4-fh = 4-fluorophenoxyacetic acid hydrazide and phen = 1,10-phenanthroline) (Figure 3) was prepared according to the published procedure in the literature [20]. Elemental analysis and UV-vis spectroscopy were carried out to verify the purity and identity of the complex.

Antimicrobials were tested in association with the copper complex for five planktonic ST strains, which were selected according to the characteristic of multidrug resistance and epidemiological distinctions. The assays were performed in three replicates, and the antimicrobial concentrations were the same as those used in the MIC test with the addition of 6.3 μg/mL (10 μM) of [Cu(4-FH)(phen)(ClO_4_)_2_] [16].

The tests of the biofilms started from the preliminary formation of the sessile structure of the five ST lineages. Initially, the cultures present in the Tryptone Soya Agar (TSA, Oxoid**^®^**, Hampshire, UK) plates were transferred to 20 mL of Tryptone Soya Broth (TSB, Oxoid**^®^**, Hampshire, UK) and incubated at 37 °C for 24 h. After growth, the bacterial suspension was standardized to OD_600_ = 0.22–0.28 and centrifuged at 5000 rpm for 10 min at 4 °C. After discarding the supernatant, the cells were washed and centrifuged twice in 0.9% sterile NaCl solution. The supernatant was discarded, and the pellet was resuspended in 0.9% NaCl solution and diluted in 10 mL of supplemented TSB in order to obtain a final count of 10^4^ CFU/mL. The technique of biofilm formation was performed according to Kudirkienė et al. (2012) [99], with modifications. Briefly, 200 µL aliquots of the bacterial suspension in TSB containing 10^4^ cells were added to 96-well plates and incubated for 48 h at 37 °C. Afterwards, the non-adherent bacteria were washed twice with 0.9% sterile NaCl solution and the biofilm formed was maintained for treatment with antibiotics.

Both the isolated effect of the antimicrobials and the association with the copper complex were evaluated in the sessile cells. The concentrations tested in biofilms were equal and up to 16 times higher than those used in the MIC test, so that the maximum values evaluated were 1024.0, 256.0, 16.0, 512.0, 512.0, 8192.0, 1024.0 and 1002.912 μg/mL for AMP, CFT, CIP, MER, COL, SUL, TET and [Cu(4-FH)(phen)(ClO_4_)_2_], respectively. In the evaluation of the synergistic effect, we used a concentration of 28.21 μg/mL (45 μM) of the copper complex to assess whether the effect was concentration- and/or strain-dependent. In addition, a 10 μL aliquot of each dilution inoculum was plated in TSA to check the bacterial growth (viability of sessile cells) of the respective dilution well. For all tests, negative controls composed of the medium without the addition of bacteria were used.

### 4.5. Scanning Electronic Microscopy (SEM)

The visualization of the biomass formed with and without [Cu(4-FH) (phen)(ClO_4_)_2_] at 100 μM was performed in an SEM with two strains of ST. The preparation of the material for analysis in the SEM was carried out according to Brown et al. (2014) [100], with modifications. Biofilms were formed in glass beads with a diameter of 5 mm, respecting the growth conditions described above. After biomass formation, the samples were fixed with 2.5% glutaraldehyde and 2.5% paraformaldehyde in 0.1 M PBS buffer (pH 7.4) overnight at 4 °C. The fixative was removed, and the samples were washed three times with PBS buffer. The beads were post-fixed with 1% osmium tetroxide for 2 h and washed three times with PBS buffer. The beads were dehydrated in a series of ethanol solutions (30, 40, 50, 60, 70, 80 and 90%, and then three times at 100%) for 15 min for each step. The samples were dried at critical drying point (CDP 030, Baltec, DE, Canonsburg, PA, USA) using liquid carbon dioxide as the transition fluid, then coated with a 20 nm thick gold layer (SCD 050, Baltec, DE, Canonsburg, PA, USA) and visualized in the SEM (VP Zeiss Supra 55 FEG SEM, operating at 20 kV).

### 4.6. Statistical Analysis

Descriptive statistics were used for data presentation. To compare the prevalence of resistance profiles and the origin of the strains, Fischer’s exact test was used with a confidence interval of 95%. The analyses regarding the synergistic effect of the use of the copper complex were carried out using the Mann–Whitney test, since the MICs did not assume a Gaussian distribution. Analyses were performed using GraphPad Prism software, version 8.0 (GraphPad Software, San Diego, CA, USA).

## 5. Conclusions

MDR was detected for all the ST strains tested, which indicated a need for alternative treatment strategies, such as the copper (II) complex.

Particularly, the high percentages of resistance/intermediate resistance to CIP and CFT are worrying as they are the drugs of choice in the treatment of the disease, as is the resistance to COL and MER, which are the final treatment alternatives for infections caused by MDR agents. The use of the copper complex was more effective compared to use of the drugs AMP/SUL/TET and AMP/SUL/TET/COL in planktonic forms and in ST biofilms, respectively. The positive synergistic effect of the use of this compound associated with all drugs tested was also observed, with variations in efficacy according to the way of life of the ST tested.

## Figures and Tables

**Figure 1 antibiotics-11-00388-f001:**
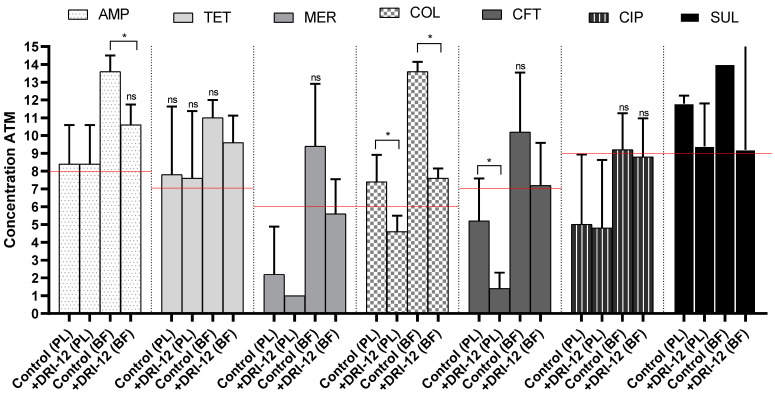
The effect of antimicrobial treatment, with and without addition of the copper complex, on the free and biofilm forms of five ST strains. The results represent means with the standard deviation (error bars) of three independent repetitions with three replicates. +DRI-12: supplementation with [Cu(4-FH)(phen)(ClO_4_)_2_] at concentrations of 10 and 45 μM, respectively, for planktonic (PL) and biofilm (BF) forms of ST. ATM concentration: antimicrobial concentration. 1–14: doubling and increasing variations in concentrations for each antimicrobial. AMP: ampicillin (1: <0.5 μg/mL, 14: >1024 μg/mL); TET: tetracycline (1: <0.5 μg/mL, 14: >1024 μg/mL); MER: meropenem (1: <0.25 μg/mL, 14: >512 μg/mL); COL: colistin (1: <0.25 μg/mL, 14: >512 μg/mL); CFT: ceftriaxone (1: <0.125 μg/mL, 14: >256 μg/mL); CIP: ciprofloxacin (1: <0.0078 μg/mL, 14: >16 μg/mL); and SUL: sulfisoxazole (1: <16 μg/mL, 12: >8192 μg/mL). Red line: cutoff point according to CLSI (2021). ns: no statistical difference in the analysis between strains for each treatment. * *p* < 0.05; using Mann–Whitney test.

**Figure 2 antibiotics-11-00388-f002:**
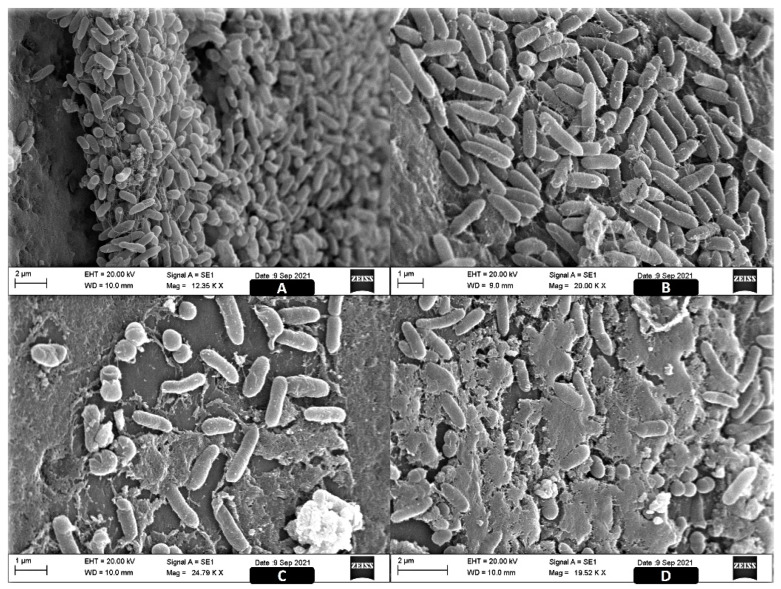
SEM images of biofilms with and without 100 μM of the copper complex treatment in two strains of ST. (**A**,**B**) control group with normal biofilm structure; (**C**) group treated with DRI-12, demonstrating the presence of a thick layer of extracellular matrix, and razing of the matrix structure and bacterial exposure; and (**D**) treatment with DRI-12, with biomass fragmentation.

**Figure 3 antibiotics-11-00388-f003:**
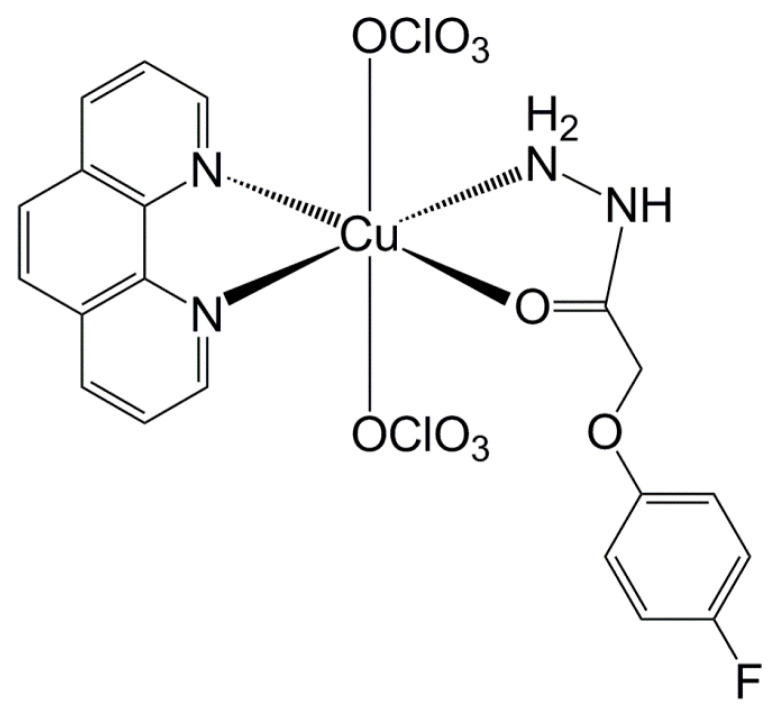
The chemical structure of [Cu(4-fh)(phen)(ClO_4_)_2_].

**Table 1 antibiotics-11-00388-t001:** The distribution of MIC (mg/L) and percentages of resistance in ST isolated from food and human samples in Brazil between 2011 and 2017.

Concentrations	AMP	TET	MER	COL	CFT	CIP	SUL	DRI-12
A	-	24	5	4	1	16	-	-
B	-	-	-	-	-	-	-	-
C	-	1	-	1	1	4	-	-
D	-	-	1	-	4	4	-	-
E	1	-	2	3	8	1	-	-
F	14	1	18	22	19	11	-	1
G	6	2	17	10	8	4	-	2
H	1	1	-	-	-	2	-	17
I	2	12	-	-	-	-	1	21
J	19	2	-	3	2	1	42	2
**R (%)**	22 (51.2)	17 (39.5)	35 (81.4)	35 (81.4)	10 (23.3)	1 (2.3)	43 (100)	-
**R + I (%)**	28 (65.1)	18 (41.8)	37 (86)	38 (88.4)	29 (67.4)	18 (41.8)	43 (100)	-
**S**	15 (34.8)	25 (58.1)	6 (13.9)	5 (11.6)	14 (32.5)	25 (58.1)	0 (0)	-
**MIC_50_**	32 mg/L	<0.5 mg/L	4 mg/L	4 mg/L	2 mg/L	0.0312 mg/L	>2048 mg/L	62.68 mg/L
**MIC_90_**	>64 mg/L	64 mg/L	8 mg/L	8 mg/L	4 mg/L	0.25 mg/L	>2048 mg/L	62.68 mg/L

A–J: doubling and increasing variations in concentrations for each antimicrobial. AMP: ampicillin (A: <0.5 mg/L, J: >64 mg/L); TET: tetracycline (A: <0.5 mg/L, J: >64 mg/L); MER: meropenem (A: <0.25 mg/L, J: >32 mg/L); COL: colistin (A: <0.25 mg/L, J: >32 mg/L); CFT: ceftriaxone (A: <0.125 mg/L, J: >16 mg/L); CIP: ciprofloxacin (A: <0.0078 mg/L, J: >1 mg/L); SUL: sulfisoxazole (A: <16 mg/L, J: >2048 mg/L); and DRI-12—copper complex [Cu(4-FH)(phen)(ClO_4_)_2_] (A: <0.49 mg/L, J: >62.68 mg/L). __ (line)—cut-off point according to CLSI 2021 [24]; dark grey highlight—resistance (R); light grey—intermediate resistance (I); white—susceptible (S); and (%)—resistance percentage.

**Table 2 antibiotics-11-00388-t002:** Genes and functions, amplification conditions, amplicon size and references used in ST assessment.

Gene	*Primers* Sequence (5′ → 3′)	Amplicon (pb)	Function	Amplification	Reference
*bla* _TEM_	CAGCGGTAAGATCCTTGAGA ACTCCCCGTCGTGTAGATAA	643	β-lactam resistance	30× (94 °C, 45 s/50 °C, 45 s/72 °C, 90 s)	[96]
*bla* _SHV_	GGCCGCGTAGGCATGATAGA CCCGGCGATTTGCTGATTTC	714	β-lactam resistance	30× (94 °C, 45 s/56 °C, 45 s/72 °C, 90 s)	[96]
*bla* _CTX-M_	TGGGTRAARTARGTSACCAGAAYCAGCGG CCCCGCTTATAGAGCAACAA	593	β-lactam resistance	30× (94 °C, 45 s/58 °C, 60 s/72 °C, 90 s)	[72]
*bla* _CMY-2_	TGGCCGTTGCCGTTATCTAC CCCGTTTTATGCACCCATGA	870	β-lactam resistance	30× (94 °C, 45 s/59 °C, 53 s/72 °C, 90 s)	[96]
*qnrA*	AGAGGATTTCTCACGCCAGG TGCCAGGCACAGATCTTGAC	580	Fluoroquinolone resistance	35× (95 °C, 60 s/54 °C, 60 s/72 °C, 90 s)	[97]
*qnrS*	GCAAGTTCATTGAACAGGGT TCTAAACCGTCGAGTTCGGCG	428	Fluoroquinolone resistance	35× (95 °C, 60 s/54 °C, 60 s/72 °C, 90 s)	[97]

**Table 3 antibiotics-11-00388-t003:** Genes identified by CT103-XL associated with bacterial resistance by producing different enzymes responsible for resistance to antimicrobials.

Carbapenamases	Β-Lactamases	AmpCs	MCR
GES	Group CTX-M-1	TEM 164C	ACC	MCR-1
GIM	Subgroup CTX-M-1	TEM 164H	ACT/MIR	MCR-2
IMP	Group CTX-M-2	TEM 164S	CMY I/MOX	
KPC	Subgroup CTX-M-3	TEM 238S	CMY II	
NDM	Group CTX-M-8	SHVwt	DHA	
OXA-23	Group CTX-M-9	SHV 238A	FOX	
OXA-24	Subgroup CTX-M-15	SHV 238S		
OXA-48	Group CTX-M-25	SHV240K		
OXA-58	Group CTX-M-32	BEL		
VIM	TEM wt	GES		
SPM	TEM 404K	PER		
		VEB

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
