# Peer review of "A Ternary Copper (II) Complex with 4-Fluorophenoxyacetic Acid Hydrazide in Combination with Antibiotics Exhibits Positive Synergistic Effect against Salmonella Typhimurium"

_antibiotics, 2022, doi:10.3390/antibiotics11030388_

Round 1
Reviewer 1 Report
Manuscript Antibiotics-1614669
The manuscript entitled “Antimicrobial resistance challenged by a copper (II) complex in Salmonella Typhimurium” presented by Guilherme Paz Monteiro and co-authors describes the emergence of Salmonella strains resistant to multiple antibiotics.
This work needs some minor changes before it can be published in the Antibiotics journal.
Results:
-Present the full names of antibiotics used in this study, in supplementary files if possible.
-Table 1 should be much easier to read, format the data and insert some more appropriate symbol for the concentration.
-Figure 1: standard deviations are too high, the assay should be repeated enough times to get a more reliable data.
-Names and descriptions of the figures should be below the figures.
-In the materials and methods you mentioned genetic study of genes associated to resistance, but haven’t written much on that subject in results section, only one paragraph. As you found that some genes encoding beta-lactamases are present in some strains, discus about that within the data you showed through the study.
-In generally some addition experiments would fulfill the study.
Discussion section has been written correctly and in details.
Author Response
Response Letter – Manuscript ID: Antibiotics -1614669
New Title: “A ternary copper(II) complex with 4-fluorophenoxyacetic acid hydrazide in combination with antibiotics exhibits positive synergistic effect against Salmonella Typhimurium”
Comments from the reviewers:
- Reviewer #1:
Limitations:
Present the full names of antibiotics used in this study, in supplementary files if possible
Author's response: The full names of the antimicrobials used in our study are available in Table 1 and Figure 1 and now also at the beginning of the supplemental files (marked in red). We thank you for your suggestion.
Table 1 should be much easier to read, format the data and insert some more appropriate symbol for the concentration
Author's response: We modified the concentration symbol by writing (concentrations). We formatted the information referring to the concentrations used for each of the antimicrobials, identified by A-J, with the description of the maximum and minimum values in the legend of the table, following the same format we did for figure 1. We adjusted the units and percentages in the last lines of the table. We believe that now the information is easier to visualize. The changes are marked in red in Table 1. We thank you for your suggestion.
Figure 1: standard deviations are too high; the assay should be repeated enough times to get a more reliable data.
Author's response: We agree that the deviations were high and point out that we performed the tests in three replicates and three independent repetitions. The effect caused by the size of the deviations refers to the fact that they represent the grouped results obtained for the 5 strains. These are strains that present multi-resistance characteristics according to the results in Table 1, but we also chose to select strains with distinct epidemiological characteristics (year, source, sample, and site - Supplementary Table 2) to be representative of the divergent characteristics they have. We obtained the same concentration results in all assays for the 5 strains (both in replicates and replicates), so we believe that if we do other replicates and replicates, probably the values will not change. The high deviations also make clear the strain-dependent characteristic, but we really prioritized to evaluate the effect considering the grouped results of the 5 strains and not specific for each strain.
Names and descriptions of the figures should be below the figures.
Author's response: We agreed with the indication and made the modification for Figures 1 and 2.
In the materials and methods, you mentioned genetic study of genes associated to resistance, but haven’t written much on that subject in results section, only one paragraph. As you found that some genes encoding beta-lactamases are present in some strains, discuss about that within the data you showed through the study.
Author's response: We have added and expanded the results regarding the above suggested in lines 126 to 133 and expanded the discussion in lines 304 to 309.
In generally some addition experiments would fulfill the study
Author's response: We agree with the issue of high deviations that you had mentioned before. But as we described, these are pooled results from the 5 strains. What we can do is show the separate results for each strain in a supplementary file, if the reviewer agrees. In this case we can make a unique graph for each strain. But we still believe that the grouped results would be more representative.
Discussion section has been written correctly and in details.
Author's response: We thank you for all your considerations.

Reviewer 2 Report
Authors
Manuscript Review Feb 20 2022
antibiotics-1614669-peer-review-v1
Article Title: Antimicrobial resistance challenged by a copper (II) complex in 2
Salmonella Typhimurium
Authors: Guilherme Paz Monteiro1, Roberta Torres de Melo1*, Micaela Guidotti Takeuchi1, Carolyne Ferreira Dumont1, Rosanne Aparecida Capanema Ribeiro1, Wendell Guerra2, Drielly Aparecida Paixão2, Fernanda Aparecida Longato dos Santos1, Dália dos Prazeres Rodrigues3, Peter Boleij4, Patrícia Giovana Hoepers5, Daise Aparecida Rossi1
Article summary:
Salmonella infection remains a very serious global health issue due to the emergence of multidrug resistant strains and its ability to form biofilms, which reduce the effectiveness of current treatment options. New strategies are needed to more effectively combat this pathogen. New treatment options for this pathogen can only be considered after thorough examination of resistance patterns and updated control strategies are tested. The authors of this manuscript characterised 43 Brazilian lineages of Salmonella Typhimurium (ST) strains, analysed the limitations in its control and proposed effective antimicrobial measures.
Review
Overall, the article was well -written and the study was sound. I have made some minor suggestions below.
Title; The title of this manuscript is slightly too vague.
Perhaps something like this: “Copper Complex in combination with certain antibiotics , provide positive synergistic effect in controlling Salmonella Typhimurium biofilm.”
Abstract:
Perhaps change the second sentence.
From This “We characterised 43 Brazilian lineages of Salmonella Typhimurium (ST) strains, analysed the limitations in its control and proposed effective antimicrobial measures.”
To: “We characterised 43 Brazilian lineages of Salmonella Typhimurium (ST) strains, characterized drug resistance patterns, tested copper (II) complex as control options, and proposed effective antimicrobial measures.”
Introduction:
Line 38. Please correct 12 to 12 hours. What did you mean? 12 to 24 hours?
Line 62-63. ……….”on the potentiation of acquisition of virulence and resistance factors.”
Perhaps slightly re write to …“on the potential acquisition of virulence and resistance factors”
Results
Line 108-110. Can the microarray data be placed in a supplemental Table or Figure?
You may need to explain this microarray in a more detail for those not familiar in the methods. It should be made clear that you are either using mRNA transcripts or DNA. It looks like you are just detecting the presence of DNA from colonies instead of mRNA transcripts.
For part of your results (section 2.2) you interchange sessile and biofilm. In figure 1, you designate column data as biofilm (BF). For consistency, it may be helpful to keep the terminology uniform such as sessile or biofilm but not both.
Discussion
Lines 197-199
“A global study carried out in 2019 showed that sulfonamides, as well as tetracyclines
(TET) and penicillins, have the highest resistance rates in poultry and swine animal pro
duction, and Brazil has emerged as one of the main emerging hotspots for the maintenance of these strains [20].”
Consider this slight change for clarity and accuracy.
“A global study carried out in 2019 showed that sulfonamides, as well as tetracyclines
(TET) and penicillins, ST isolates from poultry and swine animal production, have the highest resistance rates. Consequnetly, Brazil has emerged as one of the main emerging hotspots for the maintenance of these strains [20].”
Methods
Section 4.1
Why did the author use both PCR and Microarray? Please explain.
Section 4.4
Line 456-457
How did you determine if the ST (5 strains) was/were planktonic? How was this determined?
Section 4.5
Why were glass beads used for SEM work rather than flat glass pieces? It seems that the SEM treatments (fixatives, ppost-fix, drying, and dehydrations) would be a more uniform with flat glass piece.
Conclusions
Line 513. The sentence may need to be re written.” The results point to difficulties in treatment due to unanimous MDR”.
“MDR was detected for all the ST strains tested which indicated a need for alternative treatment strategies such as copper (II) complex.”
Author Response
Response Letter – Manuscript ID: Antibiotics -1614669
New Title: “A ternary copper(II) complex with 4-fluorophenoxyacetic acid hydrazide in combination with antibiotics exhibits positive synergistic effect against Salmonella Typhimurium”
Comments from the reviewers:
- Reviewer #2:
Limitations:
Overall, the article was well -written and the study was sound. I have made some minor suggestions below.
Author's response: We appreciate for your considerations.
Title
The title of this manuscript is slightly too vague.
Perhaps something like this: “Copper Complex in combination with certain antibiotics, provide positive synergistic effect in controlling Salmonella Typhimurium biofilm.”
Author's response: We change the title to: “A ternary copper(II) complex with 4-fluorophenoxyacetic acid hydrazide in combination with antibiotics exhibits positive synergistic effect against Salmonella Typhimurium”
Abstract:
Perhaps change the second sentence.
From This “We characterized 43 Brazilian lineages of Salmonella Typhimurium (ST) strains, analyzed the limitations in its control and proposed effective antimicrobial measures.”
To: “We characterized 43 Brazilian lineages of Salmonella Typhimurium (ST) strains, characterized drug resistance patterns, tested copper (II) complex as control options, and proposed effective antimicrobial measures.”
Author's response: We agree with the suggestion and change the text in the lines 19 and 20
Introduction:
Line 38. Please correct 12 to 12 hours. What did you mean? 12 to 24 hours?
Author's response: Sorry for the error in the writing of the text. We have changed lines 40 and 41.
Line 62-63. ……….”on the potentiation of acquisition of virulence and resistance factors.”
Perhaps slightly re write to …“on the potential acquisition of virulence and resistance factors”
Author's response: We have changed line 64. Thank you.
Results
Line 108-110. Can the microarray data be placed in a supplemental Table or Figure?
You may need to explain this microarray in a more detail for those not familiar in the methods. It should be made clear that you are either using mRNA transcripts or DNA. It looks like you are just detecting the presence of DNA from colonies instead of mRNA transcripts.
Author's response: We attach in a supplementary file the data systematized by the microarray program.
You may need to explain this microarray in a more detail for those not familiar in the methods. It should be made clear that you are either using mRNA transcripts or DNA. It looks like you are just detecting the presence of DNA from colonies instead of mRNA transcripts.
Author's response: We agree, and have added in more detail the principle of the microarray method in the methodology (Lines 448-459). Thank you for the suggestion.
For part of your results (section 2.2) you interchange sessile and biofilm. In figure 1, you designate column data as biofilm (BF). For consistency, it may be helpful to keep the terminology uniform such as sessile or biofilm but not both.
Author's response: We have changed section 2.2 to "biofilm" instead of sessile form (red in the text).
Discussion
Lines 197-199
“A global study carried out in 2019 showed that sulfonamides, as well as tetracyclines (TET) and penicillins, have the highest resistance rates in poultry and swine animal production, and Brazil has emerged as one of the main emerging hotspots for the maintenance of these strains [20].”
Consider this slight change for clarity and accuracy.
“A global study carried out in 2019 showed that sulfonamides, as well as tetracyclines (TET) and penicillins, ST isolates from poultry and swine animal production, have the highest resistance rates. Consequently, Brazil has emerged as one of the main emerging hotspots for the maintenance of these strains [20].”
Author's response: We thank you for your suggestion and have changed the text in lines 217-220.
Methods
Section 4.1
Why did the author use both PCR and Microarray? Please explain.
Author's response: We did the conventional PCR initially only as a screening and later included the microarray considering the analysis of a higher quantity of genes of interest for this bacterium. The results of both techniques were concordant. If the reviewer believes that the conventional PCR results are unnecessary because of duplication with the microarray, we can remove them.
Section 4.4
Line 456-457
How did you determine if the ST (5 strains) was/were planktonic? How was this determined?
Author's response: The planktonic form represents the free form of the bacteria, suspended in the broths used in bacterial culture. This form of the bacteria is the one used for conventional MIC tests. Obtaining this form is determined only by obtaining the bacterial suspension.
Section 4.5
Why were glass beads used for SEM work rather than flat glass pieces? It seems that the SEM treatments (fixatives, post-fix, drying, and dehydrations) would be a more uniform with flat glass piece.
Author's response: The use of glass beads for SEM testing is already standard in our laboratory, as described in our published papers (10.3389/fmicb.2017.01332; 10.3389/fmicb.2021.674147; 10.1016/j.fbio.2020.100811; 10.3390/pathogens10050581; 10.3389/fcimb.2021.535757). Using this format has never caused a problem with the quality of our samples and results. The SEM treatments reach the entire structure of the biofilm produced on the surface of the beads, which allows satisfactory analysis of the results.
Conclusions
Line 513. The sentence may need to be re written.” The results point to difficulties in treatment due to unanimous MDR”.
“MDR was detected for all the ST strains tested which indicated a need for alternative treatment strategies such as copper (II) complex.
Author's response: We thank you for your suggestion and have changed the text in lines 558-559.

Reviewer 3 Report
Dears authors
The strains analyzed are 43 strains of ST from food samples (20) and humans 359 with salmonellosis (23), I believe that the sample is low to be able to state that the use of a 516 copper complex was more effective than AMP drugs / SUL / TET and 517 AMP / SUL / TET / COL in planktonic forms and ST biofilms respectively.
The Disclosure of the data is very broad and very dispersed in particular the paragraph from line 264-276.
The images are not of orrimal quality, they must be reviewed as resolution, C (very blurry on one side) cannot be seen and the resolution factor in the rest.Above all, this work does not renew the state of the art, as there are already works on copper complexes and resistance gene factors in the literature, an in-depth study should be done by increasing the number of strains and reviewing the data at the end of the experimental work. in conclusion, in my opinion, the paper cannot be published both for the state of the art and for the consistency of the results.
Author Response
Response Letter – Manuscript ID: Antibiotics -1614669
New Title: “A ternary copper(II) complex with 4-fluorophenoxyacetic acid hydrazide in combination with antibiotics exhibits positive synergistic effect against Salmonella Typhimurium”
____________________________________________________________________
Comments from the reviewers:
- Reviewer #3:
Limitations:
The strains analyzed are 43 strains of ST from food samples (20) and humans 359 with salmonellosis (23), I believe that the sample is low to be able to state that the use of a 516 copper complex was more effective than AMP drugs / SUL / TET and 517 AMP / SUL / TET / COL in planktonic forms and ST biofilms respectively.
Author's response: The authors disagree with the reviewer. We used a significant number of strains and there are several manuscripts that can corroborate this, which used a lower quantitative number of strains for analysis of different complexes:
- Rodríguez, I.; Fernández-Vega, L.; Maser-Figueroa, A.N.; Sang, B.; González-Pagán, P.; Tinoco, A.D. Exploring Titanium(IV) Complexes as Potential Antimicrobial Compounds. Antibiotics 2022, 11, 158. https://doi.org/10.3390/antibiotics11020158
- Costa, J.P.; Sousa, S.A.; Galvão, A.M.; Mata, J.M.; Leitão, J.H.; Carvalho, M.F.N.N. Key Parameters on the Antibacterial Activity of Silver Camphor Complexes. Antibiotics 2021, 10, 135. https://doi.org/10.3390/antibiotics10020135
- O’Shaughnessy, M.; McCarron, P.; Viganor, L.; McCann, M.; Devereux, M.; Howe, O. The Antibacterial and Anti-Biofilm Activity of Metal Complexes Incorporating 3,6,9-Trioxaundecanedioate and 1,10-Phenanthroline Ligands in Clinical Isolates of Pseudomonas aeruginosa from Irish Cystic Fibrosis Patients. Antibiotics 2020, 9, 674. https://doi.org/10.3390/antibiotics9100674
- Zalevskaya, O.; Gur’eva, Y.; Kutchin, A.; Hansford, K.A. Antimicrobial and Antifungal Activities of Terpene-Derived Palladium Complexes. Antibiotics 2020, 9, 277. https://doi.org/10.3390/antibiotics9050277
The Disclosure of the data is very broad and very dispersed in particular the paragraph from line 264-276.
Author's response: We have better organized the paragraph mentioned in lines 284-297.
The images are not of orrimal quality, they must be reviewed as resolution, C (very blurry on one side) cannot be seen and the resolution factor in the rest.
Author's response: We enhance the quality of SEM images
This work does not renew the state of the art, as there are already works on copper complexes and resistance gene factors in the literature, an in-depth study should be done by increasing the number of strains and reviewing the data at the end of the experimental work
Author's response: The authors do not agree with the reviewer. In fact, there are active complexes against bacteria, which reinforces its pharmacological potential, however, there is not many studies using resistant strains. Moreover, DRI-12 is a ternary complex that contains a hydrazide ligand, namely, 4-fluorophenoxyacetic acid hydrazide. Thus, this paper is, as much as is from our knowledge, the first to describe the activity of a copper complex bearing hydrazide against several strains of Salmonella Typhimurium.
As for the resistance genes, it is important to remember that these are wild strains with phenotypic MDR characteristics, isolated from different origins and representative of various locations in Brazil. Therefore, the genotypic characterization is necessary to compare the phenotypic profile identified.
The paper cannot be published both for the state of the art and for the consistency of the results.
Author's response: The authors do not agree with the reviewer. The study was conducted with several strains of ST serovar from different origins and presenting an MDR profile. Considering that the phenomenon of resistance is a public health problem that implies the constant renewal of the therapeutic arsenal available, the need for monitoring and implementation of new control strategies is paramount. The results of this study are consistent, were statistically analyzed and were done in replicates and repetitions that ensure the reliability and robustness of the information.

Reviewer 4 Report
Over all the research work very interesting and well written the present data met to the journal quality and quantity hence i recommend the paper for publication.
Author Response
Response Letter – Manuscript ID: Antibiotics -1614669
New Title: “A ternary copper(II) complex with 4-fluorophenoxyacetic acid hydrazide in combination with antibiotics exhibits positive synergistic effect against Salmonella Typhimurium”
____________________________________________________________________
Comments from the reviewers:
- Reviewer #4:
Suggestions:
Over all the research work very interesting and well written the present data met to the journal quality and quantity hence i recommend the paper for publication.
Author's response: Thank you for recognizing our paper!

Reviewer 5 Report
Dear Authors,
Antibiotic resistance is a growing public health problem worldwide. This study is important and necessary, but novelty of the Manuscript should be improved.
The authors write „Currently, copper complexes receive much attention due to their promising results as antibacterial agents”. However, the authors mention only two published papers where the copper complex are effect on tumor cells.
Perhaps it should be mentioned why the copper (II) complex [Cu(4-FH)(phen)(ClO4)2] (4-FH = 4-fluor- 19 ophenoxyacetic acid hydrazide and phen = 1,10-phenanthroline), known as DRI-12 is better than other established agents, such as those based on nanoparticles of ZnO, Ag, TiO2 and others, doped or undoped.
Also, appropriate methods have been used to highlight working hypotheses.
The experimental results support the conclusions of the study presented in the paper. I appreciate that the bibliography is up to date.
Author Response
Response Letter – Manuscript ID: Antibiotics -1614669
New Title: “A ternary copper(II) complex with 4-fluorophenoxyacetic acid hydrazide in combination with antibiotics exhibits positive synergistic effect against Salmonella Typhimurium”
______________________________________________________________________
Comments from the reviewers:
- Reviewer #5:
Limitations:
Antibiotic resistance is a growing public health problem worldwide. This study is important and necessary, but novelty of the Manuscript should be improved.
Author's response: We have included more information regarding this complex in the introduction (lines 69-82). DRI-12 is a ternary copper complex containing a hydrazide ligand (fluorophenoxyacetic acid hydrazide), and we found that this work was the first to describe the activity of a copper (II) hydrazide-containing complex against various strains of Salmonella Typhimurium. Furthermore, our results indicate that this compound may be useful as an antibacterial agent and that hydrazide complexes should be tested for safer and more effective compounds.
The authors write “Currently, copper complexes receive much attention due to their promising results as antibacterial agents”. However, the authors mention only two published papers where the copper complex are effect on tumor cells.
Author's response: We have increased the discussion on copper complexes in the introduction and included new references.
Perhaps it should be mentioned why the copper (II) complex [Cu(4-FH)(phen)(ClO4)2] (4-FH = 4-fluor- 19 ophenoxyacetic acid hydrazide and phen = 1,10-phenanthroline), known as DRI-12 is better than other established agents, such as those based on nanoparticles of ZnO, Ag, TiO2 and others, doped or undoped.
Author's response: The chemical analysis comparing the composition of copper complexes with ZnO, Ag, TiO2 and others does not fit because they are different approaches. We altered the penultimate paragraph of the introduction (lines 69 to 82), which includes the beneficial effect of copper to human metabolism, and we also altered the discussion regarding the concentrations used in other studies with these agents by making a comparative analysis of MIC values (lines 336 to 342).
Appropriate methods have been used to highlight working hypotheses.
The experimental results support the conclusions of the study presented in the paper. I appreciate that the bibliography is up to date.
Author's response: We appreciate the recognition of the quality of the paper. Thanks for the suggestions.

Round 2
Reviewer 3 Report
The authors have provided improvements to the paper, I currently believe even if the state of the art is not changed substantially, to accept it sufficiently.